# Vaccination discourses among chiropractors, naturopaths and homeopaths: A qualitative content analysis of academic literature and Canadian organizational webpages

Eric Filice[1], Eve Dubé[2,3], Janice E. Graham[4], Noni E. MacDonald[4], Julie A. Bettinger[5], Devon Greyson[6], Shannon MacDonald[7], S. Michelle Driedger[8], Greg Kawchuk[9], Samantha B. Meyer[1]*

**1** School of Public Health and Health Systems, University of Waterloo, Waterloo, Ontario, Canada, **2** Département Anthropologie, Université Laval, Québec City, Québec, Canada, **3** Centre de recherche du CHU de Québec – Université Laval, Québec City, Québec, Canada, **4** Pediatrics (Infectious Diseases), Dalhousie University, Halifax, Nova Scotia, Canada, **5** Vaccine Evaluation Center, BC Children's Hospital and University of British Columbia, Vancouver, British Columbia, Canada, **6** Department of Communication, University of Massachusetts Amherst, Amherst, Massachusetts, United States of America, **7** Faculty of Nursing, University of Alberta, Edmonton, Alberta, Canada, **8** Department of Community Health Sciences, University of Manitoba, Winnipeg, Manitoba, Canada, **9** Faculty of Rehabilitation Medicine, University of Alberta, Edmonton, Alberta, Canada

* samantha.meyer@uwaterloo.ca

## Abstract

Vaccine hesitancy–the reluctance to receive recommended vaccination because of concerns and doubts about vaccines–is recognized as a significant threat to the success of vaccination programs and has been associated with recent major outbreaks of vaccine-preventable diseases. Moreover, the association between complementary and alternative medicine (CAM) use and vaccine hesitancy and/or refusal has been frequently reported in the literature. To date, significant gaps persist in our understanding of contemporary Canadian CAM providers' beliefs regarding vaccination and how socio-professional influences may shape their vaccine-related attitudes and behaviours. To address the latter gap, the current study aims to explore the content of professional guidelines, recommendations and other discourses among CAM providers as they concern vaccination by analyzing both academic, peer-reviewed literature and Canadian organizational webpages prepared by and/or for practicing chiropractors, naturopaths and homeopaths. In the academic literature, we identified a number of complex and diverging views on vaccination that spanned topics of effectiveness; safety; theoretical, empirical, and ethical soundness; political justifiability; and compatibility with CAM philosophy and professional boundaries. However, in its current state the CAM literature cannot be described in broad strokes as being either pro- or anti-vaccination without considering finer areas of disagreement. Compared to the academic literature, which focuses more on the conceptual and evidentiary basis of vaccination, a greater proportion of vaccine-related content on Canadian CAM organizations' webpages seems to be dedicated to offering specific directives and prescriptions to providers. Guidelines and standards of practice address a number of issues, including vaccine

**Data Availability Statement:** All relevant data are within the paper and its Supporting Information files.

**Funding:** This work was funded through a grant from the Social Sciences and Humanities Research Council of Canada (Grant #949088), awarded to ED. Website: https://www.sshrc-crsh.gc.ca/home-accueil-eng.aspx The funders had no role in study design, data collection and analysis, decision to publish, or preparation of the manuscript.

**Competing interests:** The authors have declared that no competing interests exist.

administration, counsel, education and marketing. As CAM organizations further evolve in Canada and elsewhere as part of a broader "professionalization" initiative, greater attention will need to be directed at their role in shaping providers' beliefs and practices that both support and undermine vaccine promotion efforts.

## Introduction

Vaccination, and childhood vaccines in particular, have had a major impact on improved survival in the past 100 years, second only to sanitation and clean water in number of lives saved [1]. Attaining and sustaining high vaccination coverage rates is needed for continued success. In Canada and elsewhere, an increasing number of parents are delaying and/or refusing some or all vaccines for their children, contributing to declining community protection against vaccine-preventable diseases [2–4]. Vaccine decision-making is influenced by a number of structural and environmental factors, such as accessibility, convenience and quality of vaccination services [5]. It is also, however, heavily determined by psychosocial factors, like risk perceptions [6] and social norms [7]. Vaccine hesitancy–the reluctance to receive recommended vaccination because of concerns and doubts about vaccines [8]–is recognized as a significant threat to the success of vaccination programs and has been associated with recent major outbreaks of vaccine-preventable diseases [9, 10]. The World Health Organization [11] has identified vaccine hesitancy as one of ten threats to global health.

It is well known that vaccine decisions are heavily influenced by trust in both the content and source of vaccine information [12]. Most people consider mainstream physicians and nurses to be the most trusted source for this information [13, 14]; however, some vaccine-hesitant individuals find it difficult to have an open discussion about vaccination with their physician and report feeling alienated when vaccines are discussed [15]. In contrast, vaccine-hesitant individuals reported discussions about vaccination with CAM (complementary and alternative medicine) providers (which includes, but is not limited to chiropractors, naturopaths, and homeopaths) were more in line with their ideal of a consultation (CAM providers were perceived as better listeners) and perceived the vaccination information transmitted by CAM providers to be more understandable, useful and trustworthy [16–18]. The association between CAM use and vaccine hesitancy and/or refusal has been frequently reported in the peer-reviewed literature [19–22].

As with the public, vaccine attitudes among CAM providers probably exist on a continuum, ranging from being overtly anti-vaccination, to having some doubts and/or concerns about the science of some vaccines, to being unconditionally pro-vaccination [23]. Similar to evidence-based medical healthcare providers, CAM providers' perspectives on vaccination are largely developed through socio-professional normative influences that in turn impact vaccination behaviours, beliefs regarding health and prevention, and trust in vaccine information sources [24]. Studies of Canadian naturopathic [25] and chiropractic [26] students have indicated that support for vaccination decreases with each year of CAM training. Although students generally start their training with an open mind about vaccination, both their formal education and informal socialization encourage negative beliefs about vaccination [24]. For example, a 2010 study among chiropractors showed that attitudes toward vaccination were associated with the ideology of the schools where chiropractors received their formal education; negative attitudes about vaccination were more likely to be found among graduates from schools where the training is aligned with chiropractic's historical yet outdated premise of the

primacy of the nervous system in maintaining whole-body health. According to this view, all forms of infectious disease can feasibly be remedied by spinal manipulations, thus eliminating the need for vaccines [27, 28].

To date, significant gaps persist in our understanding of contemporary Canadian CAM providers' beliefs regarding vaccination, and how socio-professional influences—including professional association guidelines and peer-reviewed literature—may shape their vaccine-related attitudes and behaviours. To address the latter gap, the current study aims to explore the content of professional guidelines, recommendations and other discourses among CAM providers as they concern vaccination by analyzing both academic, peer-reviewed literature and organizational webpages prepared by and/or for practicing chiropractors, naturopaths and homeopaths. Our intention is to identify professional norms, official recommendations and formal structures that might reflect and shape Canadian CAM providers' knowledge and practices regarding vaccination. At the outset, we wish to clarify the findings presented herein do not necessarily reflect the knowledge, attitudes and practices of CAM providers in general; we seek only to characterize the vaccine-related views promulgated by sources that putatively inform CAM practice.

## Methods

To answer our research question (How is the topic of vaccination addressed in Canadian chiropractic, homeopathic and naturopathic organization guidelines/recommendations and academic literature?), we performed a "conventional" qualitative content analysis of vaccine-related information contained in both peer-reviewed CAM-targeted academic literature and public, organizational webpages for Canadian CAM providers. Qualitative content analysis is defined as "a research method for the subjective interpretation of the content of text data through the systematic classification process of coding and identifying themes or patterns." [29, p. 1278] The "conventional" variant of content analysis which we adopt here typically involves the description of a phenomenon (in this case, vaccine knowledge, attitudes, claims, and behavioural prescriptions) without reliance on existing theory for the articulation of emergent themes [29].

### Data collection (search and screening): Peer-reviewed literature

Data were collected from the peer-reviewed, academic literature using a scoping review methodology. Similar to other review types, scoping reviews aim to survey, synthesize and describe a body of literature regarding a certain topic or series of topics. Unlike systematic reviews, which typically address narrowly-defined research questions based on precise methodological inclusion criteria (e.g. "What is the effectiveness of treatment 'x' on condition 'y' among population 'z' based on results from double-blind randomized controlled trials?"), scoping reviews answer questions that are broader in scope (e.g. "What treatments for condition 'y' have been previously reported?"). Also referred to as a kind of "evidence reconnaissance", scoping reviews can be used to describe key concepts; clarify working definitions; map the conceptual boundaries of a topic; catalogue the range of pre-existing evidence; organize evidence according to time, location, source and other dimensions; and identify research gaps in the evidence base. They can include both qualitative and quantitative data, and the "end product" is usually a broad map of the available evidence organized thematically [30]. Consistent with recommendations from Arksey and O'Malley [31] and the Joanna Briggs Institute [30], our literature search was conducted in four phases:

1. An initial, limited search of two online databases relevant to the topic (*PubMed and Web of Science*), followed by an analysis of key terms in titles, abstracts, and indexing keywords/subject headings.

2. A second search across multiple databases, including those searched in phase 1, using all identified keywords and index terms.

3. A search for additional studies in the reference lists of all articles retrieved in phase 2.

4. A hand search of key journals to identify articles that may have been missed in database and reference list searches.

We first conducted a keyword search in PubMed, then drew on MeSH (Medical Subject Headings) terms from that search to search Web of Science in order to uncover additional keywords. With the assistance of a library technician, these terms and search queries were then "translated" for use in the remaining databases, which use different search algorithms and controlled vocabulary. In sum, the following multidisciplinary databases were comprehensively searched between October 9, 2018 and November 12, 2018: *Medline* via *PubMed*, *Embase*, *PsycINFO*, *Cochrane*, *CINAHL*, and *Web of Science* (see S1 Table for search detail).

Based on the research question, the following inclusion criteria were used to determine article eligibility: (1) published in English or French; (2) published by an academic journal that uses peer review; (3) was authored by, addressed to, or contained within a journal tailored to naturopaths, homeopaths, and/or chiropractors; (4) addressed the topic of vaccination. Articles that did not meet all of the aforementioned inclusion criteria were excluded. No restrictions were placed on the date or type of publication. Sources included original research articles, review articles, commentaries, editorials, letters to the editor and book reviews. Articles were first screened and eliminated based on title and abstract, and after removing duplicates the remaining articles were then screened for full text. Additional articles were retrieved through reference list searches (phase 3) as well as hand searches of 3 key journals: *Journal of the Canadian Chiropractic Association*, *Chiropractic & Manual Therapies*, and *Homeopathy* (phase 4) (see Fig 1 for screening flowchart). Of note, an additional 19 articles were excluded during the screening process due to the following: the articles were sourced from a non-peer-reviewed periodical (n = 12); the articles were not authored by, addressed to, or contained within a journal tailored to naturopaths, homeopaths, and/or chiropractors (n = 6); the article did not address vaccination (n = 1).

## Data extraction and analysis: Peer-reviewed literature

Data were extracted from the retrieved articles and charted in MS Excel. After charting the data, all statements made with regard to vaccination were analyzed via thematic analysis. Often characterized as a "generic" qualitative method or a process shared across different analytic traditions, Braun and Clarke [32] argue thematic analysis should be considered a distinct method in its own right. In essence, it involves identifying, analyzing and reporting patterns within a dataset. Because of the inherent flexibility in thematic analysis, we feel it is important to make explicit a number of analytic decisions. First, we opted not to base our decision as to what actually constitutes a theme on any absolute frequency threshold–as will be shown, there were several sub-themes that were present only once or twice in the dataset, but nonetheless were of direct relevance to the research question. We do, however, provide frequency counts for each theme in the interest of transparency and offering additional contextual information. Second, since little is understood about CAM providers' views of vaccination, analysis was entirely inductive; themes were developed "bottom-up" from the data instead of "top-down"

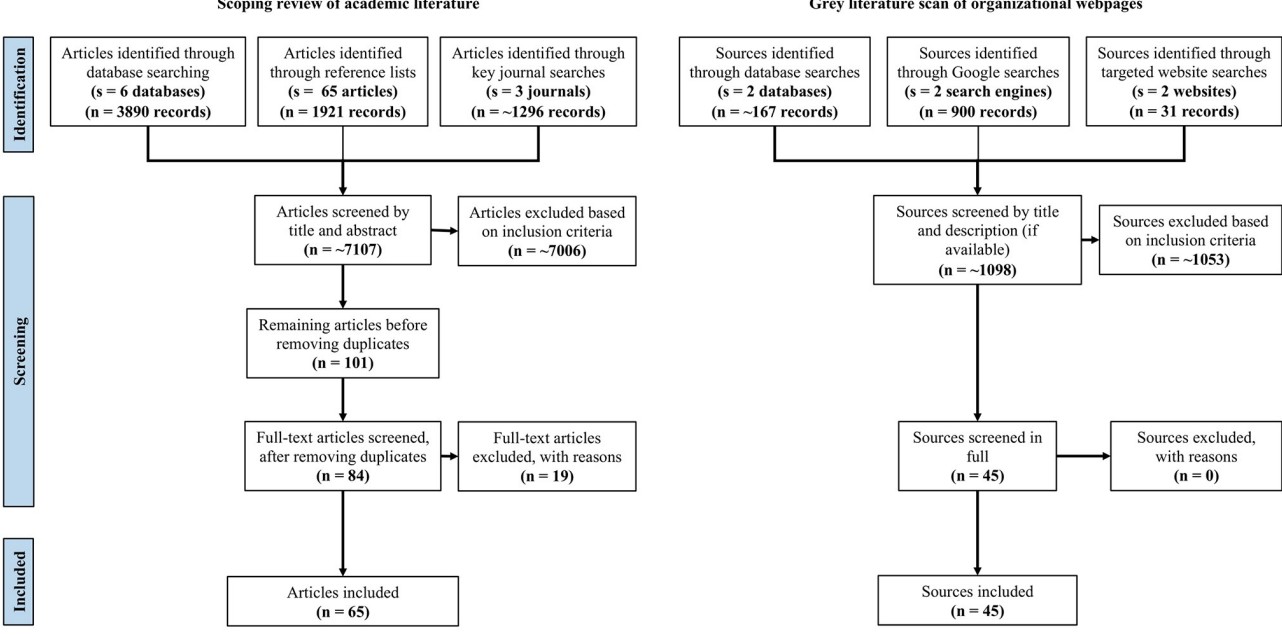

**Fig 1. Review decision flow chart for scoping review of academic literature and grey literature scan of organizational webpages.**

from an existing theoretical framework. It should also be emphasized at this point that the aims of the current study are purely descriptive–we seek only to summarize authors' claims regarding vaccination without any kind of interpretation or evaluation. As such, we do not explicitly adopt any particular theoretical or epistemological framework in our analysis (e.g. post-positivism, constructivism, etc.). Moreover, our analysis was restricted to semantic or manifest themes and avoided potential latent or interpretive themes. In other words, we focused only on the surface meanings of what was written and made no inferences concerning potential underlying assumptions, attitudes or ideologies.

Consistent with Braun and Clarke's [32] framework for thematic analysis, data were analyzed in the following sequence: 1) immersion in the data; 2) line-by-line de novo coding; 3) integrating codes into broader candidate themes; 4) refining candidate themes and developing a thematic map that illustrates connections between themes; 5) concretely defining themes and supplementing with extracted quotations. In choosing quotes to illustrate our themes, we made our best effort to provide a balanced and diverse array of viewpoints across various disciplines, authors, publication types and stances on any given issue while also ensuring the quotes selected most accurately exemplify the themes being articulated. Analysis was performed using NVivo 12.

## Data collection (search and screening): Grey literature

Whereas data were retrieved from the peer-reviewed CAM literature via a scoping review approach, for Canadian professional organization webpages we adopted a method used for scans of grey literature. Our search protocol was based on a template proposed by Godin et al. [33], which was developed and applied to a systematic review of the grey literature pertaining to guidelines for school-based breakfast programs in Canada. Based on Godin et al.'s [33] recommendations, we triangulated sources acquired from three different search strategies in

order to minimize the risk of overlooking relevant sources: grey literature databases, Google internet searches, and targeted website searching. Searches were performed from March 2[nd] to 24[th], 2019. Databases that compile and index grey literature, including the *Canadian Research Index* and *Public Health Grey Literature Database*, were searched using controlled vocabulary, and additional grey literature sources were retrieved by regular and advanced internet searches using the Google search engine. We sorted all Google search results in order of descending relevance and reviewed only the first ten pages (100 results) for each search input. To minimize the influence of personal browser histories and cookies on Google search results, all searches were performed through Google Chrome's "incognito mode". Finally, we hand searched a number of webpages that indexed other organizational webpages, including those from the Canadian Chiropractic Association and the Canadian Association of Naturopathic Doctors.

Because of the greater variability of information sources, the eligibility criteria used in the scoping review methods had to be modified slightly to suit a grey literature search. Sources were only included in the review if they met *all* of the following inclusion criteria: (1) published by a government entity at the municipal, provincial or federal level; nongovernmental organization; regulatory body; professional organization or equivalent within Canada; (2) available in English or French; (3) is the most current version of a published document; (4) intended for use by registered naturopaths, homeopaths or chiropractors. Since grey literature sources rarely have an abstract, we instead based our initial screen on title, organization name, abstract, executive summary or table of contents–whatever was available (see Fig 1). Like the scoping review, data from the retrieved sources were extracted into a spreadsheet template and analyzed via inductive thematic analysis.

## Results

The order in which we present our findings reflects the assumption that regulations and/or recommendations posed by professional organizations shape individual providers' practice, but not necessarily in uniform ways. We first detail these professional regulations and recommendations before describing vaccine-related information reported by authors in the CAM literature, thus offering readers an indication of the variability in how individual CAM providers may endorse or comply with regulatory influences.

### Characteristics of included professional organizational webpages

Of the ~1098 sources screened through searches of the grey literature, 45 webpages met the inclusion criteria and were included in the final review (see Fig 1). Of these, 34 were curated by professional organizations or associations, 10 by regulatory or licensing colleges, and 1 by research organizations. Most webpages were dedicated to chiropractic (44.4%) or naturopathy (40.0%), with homeopathy comprising a relatively smaller share (15.6%). Over one-quarter (28.9%) of organizational webpages indicated they included all of Canada within their geographic scope of coverage or jurisdiction; others exclusively serve certain provinces, and one source includes all of North America in its coverage. 86.6% of webpages were last updated in 2017 or later, with 44.4% of webpages being updated in 2019 (see Table 1 and S2 Table).

### Themes from organizational webpages

**Chiropractic organizations.**   Of the 20 Canadian chiropractic organizational webpages surveyed, the majority make no explicit reference to vaccination or immunization (n = 13). Those that do address vaccination generally note that:

**Table 1. Overview of characteristics of professional organization webpages.**

| Characteristic | n (%) |
|---|---|
| Source type | |
| Association/organization webpage | 34 (75.6%) |
| Regulatory/licensing college webpage | 10 (22.2%) |
| Research organization webpage | 1 (2.2%) |
| Profession | |
| Chiropractic | 20 (44.4%) |
| Naturopathy | 18 (40.0%) |
| Homeopathy | 7 (15.6%) |
| Geographic coverage/jurisdiction | |
| Canada | 13 (28.9%) |
| Ontario | 6 (13.3%) |
| British Columbia | 6 (13.3%) |
| Québec | 5 (11.1%) |
| Manitoba | 3 (6.7%) |
| Alberta | 2 (4.4%) |
| New Brunswick | 2 (4.4%) |
| Nova Scotia | 2 (4.4%) |
| Prince Edward Island | 2 (4.4%) |
| Saskatchewan | 2 (4.4%) |
| Newfoundland | 1 (2.2%) |
| North America | 1 (2.2%) |
| Date last updated | |
| 2019 | 20 (44.4%) |
| 2018 | 15 (33.3%) |
| 2017 | 4 (8.9%) |
| 2016 | 2 (4.4%) |
| 2014 | 1 (2.2%) |
| 2012 | 1 (2.2%) |
| Unknown | 2 (4.4%) |

- Vaccination is an established public health practice for the prevention of infectious disease (n = 5);

- Vaccination is not within the scope of chiropractic practice (n = 7);

- Health professionals whose scope of practice includes vaccination are appropriate sources for counsel on the subject (n = 4).

These points are distilled in the Canadian Chiropractic Association's 2015 [34] position statement, which was also adopted verbatim by several provincial organizations, including the Manitoba [35], Ontario [36] and British Columbia Chiropractic Associations [37]. It stipulates the following:

> The Canadian Chiropractic Association recognizes that vaccination and immunization are established public health practices in the prevention of infectious diseases. Vaccination is not within the scope of chiropractic practice. The appropriate sources for patient consultation and education regarding vaccination and immunization are public health authorities and health professionals with a scope of practice that includes vaccination.

Other organizations, including the Colleges of Chiropractors of Ontario [38], British Columbia [39] and Alberta [40], issue a series of directives to registered members with regard to vaccination based on the aforementioned principles. Practicing chiropractors are expected to:

- Inform clients vaccination is outside their scope of practice (n = 1);

- Advise clients to discuss vaccination with health professionals whose scope of practice includes vaccination (n = 2);

- Encourage clients to make informed decisions regarding their health care (n = 1).

These organizations simultaneously prohibit or expressly discourage members from the following:

- Expressing any personal opinions on vaccination to clients or the public (n = 1);

- Offering counsel about vaccinations to clients (n = 3);

- Conducting seminars about vaccination (n = 1);

- Providing information concerning vaccination on one's personal website (n = 1);

- Supplying information related to vaccination to one's own clinic or any venue where one practices (n = 1).

## Naturopathic organizations

Like chiropractic organizations, the majority of webpages for Canadian naturopathic organizations do not address vaccination (n = 13 of 18). Those that do, like the Colleges of Naturopaths of Alberta [41], Ontario [42] and British Columbia [43], make similar claims as chiropractic organizations regarding providers' scope of practice vis-à-vis vaccination. They also note that:

- Both benefits and risks are inherent to vaccination, but the benefits significantly outweigh the risks (n = 1);

- There exist no known alternatives to vaccination that produce equal or similar results (n = 2);

- Patients have a right to accept or refuse any form of medical treatment, including vaccination (n = 2).

In an editorial titled *Ten Healthy Brain Tips*, The Ontario Association of Naturopathic Doctors [44] implied vaccines are a common source of mercury, which results in brain disease.

In addition to the aforementioned directives from chiropractic practice guidelines, naturopathic organizations stipulate providers must:

- Inform clients they are not authorized to prescribe or administer vaccines (n = 1), unless officially certified by the province to do so (n = 2);

- Inform clients there is inherent risk of infection if they forego vaccination (n = 1);

- Counsel patients on the benefits or risks of vaccinations, or lack thereof, so that they may make an informed decision (n = 2);

- Familiarize oneself with the provincial routine immunization schedule, and accurately relay the immunization schedule to clients (n = 1);

- Inform clients their choices concerning vaccination will not impact the client-provider relationship (n = 2).

  Additional restrictions on practice include:

- Administering vaccines (n = 2), unless providers are officially certified by the province to do so (n = 2);

- Advising against vaccines unless they have a sound and properly documented medical reason for doing so (n = 1);

- Providing clients or the general public with materials that advise against or address the harms/risks of vaccination (n = 1);

- Including 'anti-vaccination' content or since-discredited vaccine-related evidence in advertising or marketing materials (n = 2);

- Informing clients vaccines are a likely cause of autism, or offering or advertising CEASE (Complete Elimination of Autism Spectrum Expression) therapy (n = 1);

- Advertising, offering or recommending alternatives to vaccination (n = 3).

## Homeopathic organizations

4 of the 7 Canadian homeopathic organizations address vaccination on their webpages. Apart from the aforementioned statements made by chiropractic and naturopathic organizations, organizations like the College of Homeopaths of Ontario [45] noted:

- One could reasonably expect advising clients against vaccination to result in bodily harm (n = 1);

- Nosodes are not equivalent to vaccines in biomechanics or effectiveness (n = 1).

The Manitoba Homeopathic Organization [46] posted an editorial titled "Exposing Fraudulent Arguments That Favor Forced Mass Vaccination" that contains a number of vaccine-related claims, including:

- Certain vaccines cannot prevent infectious disease transmission, either because they are not designed to do so or because the diseases they target are non-communicable;

- The claim that vaccine-related adverse events are uncommon is unsupported by evidence;

- The low severity and susceptibility of certain vaccine-preventable illnesses makes vaccination difficult to justify in light of its adverse effects;

- Because the risks of adverse events from vaccination is so high, vaccination should remain voluntary;

- To prohibit school admission for not being vaccinated against certain diseases would constitute discrimination.

The BC Association of Homeopaths [47] also provided a link under their "resources" page to Vaccine Risk Awareness Network, operated by Vaccine Choice Canada, a non-profit organization that recently came under public scrutiny for paying to erect over 50 vaccine-critical billboards in Toronto, Ontario in February 2019 [48].

## Characteristics of included peer-reviewed literature

Of the 3890 articles initially screened through database searching, 52 were included in the final review. An additional 10 articles were included from reference list searches as well as 3 articles retrieved from hand searches of key journals. In total, 65 articles met the inclusion criteria and were included in the final review of peer-reviewed literature (see Fig 1). Of these, 28 were review articles (which ranged considerably in the degree to which methods were specified); 23 were commentaries, editorials or opinion articles; 7 were original research articles that involved primary data collection of some kind; 5 were letters to the editor; and 2 were book reviews. Over half (56.9%) of all included articles were contained within journals specializing in homeopathy, while approximately one-quarter (26.1%) were sourced from chiropractic journals. Other subject areas from which articles were sourced include general medicine (6.2%), vaccines (1.5%), and bio- or medical ethics (1.5%). The majority of articles were contained in journals published within the United States (40%) or Germany (32.3%). Articles' first authors most often resided in the United States (44.6%), Australia (16.9%), the United Kingdom (7.7%), Canada (6.1%), India (6.1%), and Brazil (4.6%). Nearly half (46.1%) of all articles were published in 2010 and beyond, 46.2% were published between 2000 and 2009, and only 7.7% of articles were published prior to 2000 (see Table 2 and S3 Table).

We also sought to examine the common sources of evidence cited by the included articles. After randomly selecting 30 of the 65 articles and screening at random up to 20 sources included in each of their reference lists, we found an average 8.95 sources (44.75%) originated from peer-reviewed CAM journals, non-peer-reviewed CAM periodicals, books dedicated to CAM, or CAM conference proceedings. A slightly smaller proportion originated from scientific journals unrelated to CAM (7.98 per 20 sources, or 39.88%). Other less frequently cited sources included news articles, historical documents, government publications, self-published manuscripts, internet webpages and personal correspondence.

## Themes from peer-reviewed literature

**Vaccine effectiveness.** 27 of the 65 surveyed academic articles made claims regarding vaccine effectiveness. Many underlined that vaccines are not 100% effective, but nonetheless have been indispensable in improving and safeguarding population health (n = 17). For instance, Teixeira [49, p. 215] notes,

> Regarding the controversial subject of vaccines, homeopaths cannot deny the immense legacy they have brought to collective health, eradicating a series of epidemics which continue to ravage humanity in underdeveloped regions, that lack basic health care.
>
> [article type: narrative review with unspecified methods]

Other sources suggest vaccination is generally ineffective in preventing transmission of communicable diseases (n = 12). Of these, some refer to cases of disease outbreaks in populations with high vaccination rates (n = 1), or the time lag between mass vaccine implementation and observable epidemiologic changes, to underscore their alleged ineffectiveness (n = 1). Other authors express skepticism towards claims that population-level declines in infectious disease prevalence are attributable to vaccination, arguing epidemiologic shifts commonly associated with vaccination are in fact due to environmental changes, like improved sanitation, hygiene, and nutrition (n = 4). Morrel [50, p. 108] submits:

> The data presented by Bedford and Elliman do not conclusively show that vaccination caused the decline of infectious diseases. Diphtheria, tuberculosis, and pertussis were

**Table 2. Overview of characteristics of peer-reviewed literature.**

| Characteristic | n (%) |
|---|---|
| Article type | |
| Review article | 28 (43.1%) |
| Commentary/editorial/opinion article | 23 (35.4%) |
| Original research article | 7 (10.8%) |
| Letter to the editor | 5 (7.7%) |
| Book review | 2 (3.1%) |
| Country of origin of publisher | |
| United States | 26 (40.0%) |
| Germany | 21 (32.3%) |
| India | 5 (7.7%) |
| United Kingdom | 5 (7.7%) |
| Canada | 5 (7.7%) |
| Brazil | 2 (3.1%) |
| Netherlands | 1 (1.5%) |
| Country of residence for first author at time of publication | |
| United States | 29 (44.6%) |
| Australia | 11 (16.9%) |
| United Kingdom | 5 (7.7%) |
| Canada | 4 (6.1%) |
| India | 4 (6.1%) |
| Brazil | 3 (4.6%) |
| Italy | 2 (3.1%) |
| Netherlands | 1 (1.5%) |
| France | 1 (1.5%) |
| Argentina | 1 (1.5%) |
| New Zealand | 1 (1.5%) |
| Greece | 1 (1.5%) |
| Germany | 1 (1.5%) |
| Unknown | 1 (1.5%) |
| Date of publication | |
| 1990–1994 | 3 (4.6%) |
| 1995–1999 | 2 (3.1%) |
| 2000–2004 | 12 (18.5%) |
| 2005–2009 | 18 (27.7%) |
| 2010–2014 | 19 (29.2%) |
| 2015–2019 | 11 (16.9%) |
| Subject area of journal in which article is published | |
| Homeopathy | 37 (56.9%) |
| Chiropractic and/or manual therapies | 17 (26.1%) |
| Complementary and alternative medicine | 5 (7.7%) |
| General medicine | 4 (6.2%) |
| Vaccines | 1 (1.5%) |
| Bio- and medical ethics | 1 (1.5%) |

virtually extinct before vaccines were introduced. American and British data show similar patterns. More likely causes are improved water supply, sanitation, adequate food supply, and birth control.

[letter to the editor]

Others suggest that the apparent improvements in population health are in actuality a result of changes in diagnostic criteria which either increase the 'threshold' of the clinical disease state or rework disease classifications altogether (n = 2). Some also argue reductions in disease prevalence are simply indicative of a waning phase in larger natural cyclic fluctuations of disease-causing agents (n = 4). Irrespective of their proclaimed ability to prevent the acquisition and transmission of infectious disease, it was stressed that vaccines are ineffective in ameliorating environmental conditions that promote reproduction of infectious agents and generate disease vulnerability (n = 1).

## Vaccine safety

39 articles made claims related to vaccine safety. Vaccines were purported to be a primary cause for a number of acute and chronic conditions (n = 24), including chronic autoimmune and allergic diseases (n = 9), autism (n = 9), attention deficit disorder (n = 1), obsessive-compulsive disorder (n = 1), behavioural problems (n = 3), learning disabilities (n = 2), 'criminality' (n = 1), scurvy (n = 1), diabetes (n = 1), a variety of cancers (n = 2), recurrent miscarriage (n = 1), brain damage (n = 1), temporary coma (n = 1), physical disability (n = 1), AIDS (n = 3), and death (n = 5). The burden of vaccine-related illness was suggested to be so great that it has resulted in dramatic shifts in the epidemiologic profile of entire populations (n = 5). Moskowitz [51, p. 19] argues:

Far from being inexpensive, let alone an unmixed blessing for the public health, vaccination represents an enormous hidden cost and risk factor to the medical system as a whole, a hugely expensive and dangerous experiment that has already overburdened and sickened the population, and will undoubtedly continue to do so. Only our blind, quasi-religious faith in it, unique in all the world, will suffice to explain the scandal that the United States spends so exorbitantly on medical care, yet lags so far behind all other developed countries in every standard health measure.

[commentary/editorial/opinion]

A number of mechanistic explanations are offered for how vaccinations actually result in illness (n = 15). Some suggest that it is not improper prescription or administration but the composition of vaccines themselves that result in toxic effects (n = 2). According to this view, there are no 'right' and 'wrong' ways to administer vaccines, as all vaccines are inherently dangerous. In the same vein, it is argued that vaccine-related illness is not isolated to specific vaccines, but are a result of the general process of vaccination. Others claim vaccine pathogenicity is due either to the deliberate use of chemical additives (n = 5), like thimerosal, mercury, aluminum, and formaldehyde, or unforeseen issues in the manufacturing process, such as biologic contamination (n = 3), which are usually the result of poor quality control (n = 1). Conversely, some assert the issue is not with vaccine composition, but improper administration (n = 4). Vaccine-related illness is posited to result from administering vaccines too early in the lifespan (n = 1), administering vaccines too frequently (n = 1), or administering too many vaccines at the same time (n = 4).

On the other hand, a portion of authors tended to qualify or under-emphasize alleged links between vaccines and side effects (n = 9). Some note that vaccine-related adverse events are infrequent and the more common issues, like fever and soreness at the injection site, are rarely serious (n = 3). When problems do occur, it is argued they almost always result in a full recovery (n = 1). While it is acknowledged all interventions have a risk component, some assert vaccines do not carry any greater risk than the average medical procedure (n = 2). As well, like other medical procedures, vaccination becomes ever more safe with continued improvements to manufacturing, administration and evaluation (n = 1). Others scrutinize claims from their colleagues that vaccines are linked with certain illnesses, often pointing to the methodological and empirical limitations of the evidence these authors cite (n = 6). Ferrance [52, p. 171] notes, for instance:

> Both SIDS and autism are tragic, and any potential cause needs to be investigated fully. That has been done in the case of vaccines, and quite simply, the data is not there to support a causal link, no matter how strenuously the National Vaccine Information Center might argue in their literature and on their webpage.

[commentary/editorial/opinion]

## Medical vaccine research

A number of claims concerning vaccine safety and effectiveness are premised on the quality and appropriateness of current research (n = 13). For example, some sources critique the methodology of research demonstrating effectiveness and safety of vaccines (n = 5). Some of the cited shortcomings include the inability to infer causation from correlational data (n = 2); the paucity of longitudinal evidence in observational studies (n = 5); the disproportionate focus on quantitative data collection and analysis that obscures the unique, qualitative features of vaccine reaction (n = 1); the limited external validity of randomized controlled trials (n = 1); the focus on proportional values over crude rates in epidemiological figures, which obscure the absolute impact of vaccine damage (n = 1); and the subjectivity inherent in establishing inclusion criteria for systematic reviews, which can result in biased interpretations of quantitative data (n = 1). Some posit the current composition of the literature, which overwhelmingly contains evidence that validates rather than falsifies claims of vaccine safety and effectiveness, is a result of systemic editorial bias (n = 2). Vaccine researchers are also accused of various forms of academic misconduct, including unduly manipulating or falsifying data (n = 2), failing to properly disclose financial conflict of interest (n = 2), and obfuscating possible alternative interpretations of statistical data through partial presentation (n = 1). Ferrance [53] is more defensive of the methodological rigour and integrity of vaccine research, asserting researchers have, as a whole, been impartial and honest about the risks and benefits of vaccination, and that research is continually increasing in both quality and quantity.

It is also speculated that a number of institutional powers are directly invested in managing public awareness of vaccine-related adverse events (n = 11). Providers of evidence-based medicine are accused of routinely dismissing patients' concerns of vaccine damage (n = 2), as well as under-reporting adverse events to state-level monitoring and surveillance systems (n = 4). Some posit providers pharmacologically suppress vaccine-related symptoms to avert detection by patients (n = 1). Even when they acquire data from monitoring systems that would draw concern, major public health agencies like the Food and Drug Administration (FDA) and the Centres for Disease Control (CDC), influenced by competing interests from vaccine manufacturers (n = 2), allegedly conspire to systematically cover up the extent of vaccine damage by

obfuscating and expunging confirmatory evidence (n = 2). Kent and Gentempo [54, pp. 15–16] employ the following historical example to support this argument:

> J. Anthony Morris, one time head of influenza control in the U.S., warned his superiors in the federal government that the vaccine was dangerous and probably ineffective. When they refused to act, he went directly to the media. Morris advised the public that the vaccine was unsafe, and an epidemic was unlikely. As a result, he was fired from his position at the Food and Drug Administration. His experimental animals, representing years of research, were destroyed. Publication of his findings were blocked by his superiors.
>
> [commentary/editorial/opinion]

## Alternatives to vaccination

Several strategies besides vaccination are positioned as being appropriate for augmenting immune function, particularly for vaccine-hesitant clients, including homeopathy (n = 4); chiropractic (n = 1); environmental interventions, like sanitation measures and improved access to nutritious foods (n = 1); and lifestyle-related preventive strategies, like diet, exercise, and stress prevention (n = 3). Conversely, it is also argued that while vaccination is not perfectly effective or safe, it is superior to alternatives (n = 1).

The use of homeopathic therapies for disease prevention, also referred to as homeopathic prophylaxis or homeoprophylaxis, is positioned by several authors as a viable alternative to immunization via vaccination (n = 7). Some, like Golden [55, p. 123], claim the effectiveness and safety of homeoprophylaxis rivals, if not exceeds that of vaccination (n = 12):

> This fact highlights the potential value of homeoprophylaxis (homoeopathic immunization) in the public health debate. With HP [homeoprophylaxis], parents can obtain a significant level of protection against diseases that concern them (around 90%—comparable to vaccine effectiveness), without any risk of toxic damage. My latest research has also demonstrated that appropriate HP does not cause any long-term energetic/constitutional damage.
>
> [narrative review with unspecified methods]

Other benefits of homeoprophylaxis beyond safety and effectiveness are emphasized, including affordability, flexibility and rapidity in developing new solutions in response to emerging conditions, as well as ease in its generally non-invasive administration (n = 2). Biomechanical explanations for homeoprophylaxis' purported superior effectiveness and safety over vaccination suggest that unlike vaccination, which reportedly forces the immune system into a state of chronic generalized response, homeoprophylaxis gently enables the body to 'educate' itself on how to heal from infectious disease (n = 3). While some argue the effectiveness and/or safety of homeoprophylaxis has been supported by evidence (n = 8), others note the literature is more nascent and equivocal than supporters would care to acknowledge (n = 4).

## Ethical and political perspectives on vaccination

Vaccination is also discussed in ethical and political terms, particularly with regard to mandatory vaccination (n = 9). Some employ a utilitarian framework in defense of these programs, arguing mandatory vaccination, like seatbelts and smoking bylaws, maximizes collective well-being(n = 3). Others claim programs of this type infringe upon individual civil liberties and

are part and parcel of an excessively authoritarian government regime (n = 5). These same authors assert that debates around vaccine safety and effectiveness remain tangential to the more pressing issue of individual autonomy–regardless of its collective benefits, the decision on whether to vaccinate should rest with the individual and not the state. By extension, critiques of public anti-vaccination views are framed as an attack on individuals' right to free speech (n = 1). Vernon and Kent [56, p. 43] argue restricting individual rights is only justifiable insofar as noncompliance with a particular regulatory project poses direct harm to others, and that certain vaccine-preventable illnesses are not adequately injurious, widespread or readily transmissible to warrant mandatory vaccination. They explain:

> Experts have argued that in light of the low mortality and morbidity associated with chicken pox, as well as the unknown long-term efficacy of the varicella vaccine, that it may fail the definition of "serious consequences if transmitted." The same argument is made by medical ethicists with regard to the routine immunization against hepatitis B. Because this is a disease that is spread through sexual contact and intravenous drug use, and has a potential for serious adverse reactions, some experts argue that this vaccine should be limited to high-risk populations and should not be given on a routine basis.

[narrative review with unspecified methods]

## Homeopathic and chiropractic philosophy

A number of vaccine-related arguments involve discussion of whether vaccination and the germ theory of disease on which the procedure is based are commensurate with traditional chiropractic and homeopathic philosophies as articulated by foundational works in each field (n = 10). Opinion seems divided on whether homeopathy's principle forbearer, Samuel Hahnemann (1755–1843), was partial to vaccination. Some claim Hahnemann was explicit about the positive impact of the smallpox vaccine on population health (n = 2), while others assert Hahnemann condemned vaccines for their toxicity and incompatibility with the homeopathic directive to administer individualized treatment (n = 4). The topic is similarly polarizing in the field of chiropractic, with some noting the discipline's founder, Daniel David Palmer, was in no uncertain terms opposed to vaccination (n = 3), and another positing vaccines' mode of action does not necessarily conflict with the original principles he set forth regarding appropriate methods for treatment and prevention (n = 1).

As this debate continues between some authors, others question whether the biomedical principles of vaccination need be compatible with traditional homeopathic/chiropractic philosophy to accept their importance in safeguarding public health (n = 3). These authors argue it is essential to distinguish between taking philosophical/ideological and evidence-based positions on vaccination, the latter suggested to be more productive for determining how best to protect population health. Cooperstein [57, p. 28] articulates this point as follows:

> Again, I argue against seeing the vaccination question as a matter of philosophy, but if I must, I am forced to conclude that vaccination procedures are very much in keeping with our philosophy. I must reiterate that I am neither advocating nor attacking vaccination. Rather, I am respectfully asking that doctors of chiropractic view this as a scientific matter and a question of public health, not a philosophical matter.

[commentary/editorial/opinion]

## Professional boundaries of chiropractic and homeopathy

Arguments concerning providers' obligation to inform themselves and their patients on vaccination are periodically premised on assumptions of their scope of practice (n = 6). Some authors note that chiropractors are not trained or authorized to administer vaccines; nevertheless, as representatives of ancillary health care services and, for many patients, the primary point of contact with the health care system, it is argued they must be capable of providing reliable consult on contemporary health issues, including vaccination (n = 3). However, claims that providers should discuss vaccination do not imply their explicit goal should be to increase or decrease uptake, per se. Johnson et al. [58, p. 497] specifically assert chiropractors have a responsibility to complement evidence-based medicine in vaccine promotion:

> Health care providers are in a position to advise and influence the decision-making process of their patients, the public. Although medical practitioners and nurses are uniquely positioned to directly participate in the immunization process, DCs [Doctors of Chiropractic], as representatives of a large alternative and complementary primary contact health care profession, have the opportunity to contribute to immunization programs through the process of responding to patient queries about issues related to vaccines with evidence-supported information.
>
> [narrative review]

Though some claim chiropractic and homeopathic practice should be based on sound evidence, even if that means adopting treatment and prevention strategies from evidence-based medicine (n = 2), it is also suggested CAM providers have a moral obligation to resist biomedical hegemony by rejecting its techniques (n = 1).

## Views on vaccine-related attitudes and practices

A number of authors also offer their views on the vaccine-related attitudes and practices they observe among the general public and within their own professional circles (n = 19). With regard to critical perspectives of vaccination, it was argued that: they are not necessarily irresponsible nor do they pose a direct hazard to others (n = 1); they are motivated primarily by concerns of safety, reciprocity and full disclosure in the medical encounter (n = 1); they are rooted in a desire to minimize one's own or their children's risk of harm from infectious disease (n = 4); they emerge as clients become increasingly knowledgeable about vaccines' purported ineffectiveness and adverse effects (n = 3); and they involve engaging critically with issues of biomedical hegemony, medical paternalism and behavioural regulation vis-à-vis disciplinary power (n = 2). Saltzman [59, p. 37] exemplifies the last premise when she posits:

> It is our society's blind allegiance to authority or "the experts" on many different levels, whether it's parents' unquestioning faith in their doctors, their doctors' trust in the CDC, or the public's deference to the medical establishment/ pharmaceutical industry in general, that is literally killing us.
>
> [commentary/editorial/opinion]

Conversely, others note that these positions: are socially, ethically or professionally irresponsible (n = 2); contradict leading evidence (n = 4) and are based on belief systems that necessitate unscientific reasoning (n = 3); threaten public health, particularly with the risk they pose of reintroducing previously eradicated infectious diseases to the population (n = 3);

undermine the legitimacy of complementary and alternative medicine (n = 2) and impede the health promotion objectives of allied health disciplines (n = 1); are a result of information literacy deficits (n = 1); and stem from an unreflexive disavowal of any interventions associated with evidence-based medicine (n = 2).

### Recommendations concerning vaccination

Some articles offer explicit recommendations concerning vaccination to both clients and fellow service providers (n = 21). Two sources outright advise readers against all forms of vaccination (n = 2). Master [60, p. 4] notes:

> If I weigh all this then I can't see any reason why I could recommend vaccinations. I have refused vaccinations for many years and if a patient absolutely wants to get vaccinated then he has to resort to another person.

> [commentary/editorial/opinion]

Others recommend receiving only certain vaccines or getting vaccinated only under certain conditions. Parents are encouraged to avoid combined vaccines (vaccines that immunize against multiple antigens simultaneously) and to only vaccinate their children after 18 months (n = 1). Patients are also advised to consider chiropractic services, homeopathic services, and lifestyle modifications as supplements or alternatives to vaccination, which may amplify its protective benefits, or, in the case of total substitution, offer protection without the risk of vaccine damage (n = 2). Finally, patients are advised to consider seeking legal exemption from mandatory vaccination programs (n = 1).

Authors advise their colleagues in CAM to: first consider whether vaccination is necessary before recommending it to clients (n = 1); not feel forced to advocate vaccination so as to achieve professional legitimation under the auspices of biomedical hegemony (n = 1); advise homeoprophylaxis alongside or as a standalone alternative to vaccination (n = 5); and treat vaccine damage with homeopathic preparations (n = 1). Other sources encourage providers to: explicitly promote vaccination to patients (n = 3); avoid advising clients against vaccination (n = 1); avoid advising clients to violate compulsory vaccination laws (n = 1); offer counsel for vaccination in adherence with recommendations and provisions from state laws, public health authorities, and evidence-based medicine (n = 3); keep up-to-date on the current state of vaccine research (n = 1); and present vaccine-related information in a way that is consistent with leading evidence and free from personal bias (n = 4). Khorsan and colleagues [61, p. 501] assert:

> In circumstances where chiropractors or other health care providers are consulted about vaccination, they have a professional responsibility to provide current, accurate, unbiased (both positive and negative) information based on sound scientific evidence. This information is necessary in supporting the patients' ability to make a truly informed choice.

> [commentary/editorial/opinion]

## Discussion

In an effort to elucidate how formal structures and professional norms could potentially influence CAM providers' views toward vaccination, the purpose of the current study was to explore how the topic of vaccination is discussed in professional guidelines/recommendations and academic literature in the fields of chiropractic, naturopathy and homeopathy.

A content analysis of articles in peer-reviewed CAM journals revealed a number of complex and diverging views on vaccination that spanned topics of effectiveness; safety; theoretical, empirical and ethical soundness; political justifiability; and compatibility with CAM philosophy and professional boundaries. However, few authors seem to make explicit, unqualified statements in support or rejection of vaccination. The CAM literature cannot be described in broad strokes as being either pro- or anti-vaccination without considering finer areas of disagreement. That being said, certain vaccine-related claims tend to cluster in patterned ways—those authors who attest to vaccines' safety are also likely to vouch for their effectiveness and social-political defensibility, for example. In this sense, while vaccine attitudes among authors in CAM journals are on a spectrum, they appear to lean towards a bimodal distribution in that most make a case either in favour of or against vaccination—few authors seem to be ambivalent or tentative on their position, nor do many seek a "compromise" between opposing camps or advocate for further inquiry before taking a definitive stance. Moreover, the degree of polarization appears to vary to some extent between disciplines. Though articles that are supportive of vaccination originate from both the chiropractic and homeopathic literature, they seem to be more prevalent in the former.

Compared to the academic literature, which focuses more on the conceptual and evidentiary basis of vaccination, a greater proportion of vaccine-related content on Canadian CAM organizations' webpages seems to be dedicated to offering specific directives and prescriptions to providers. Guidelines and standards of practice address a number of issues, including vaccine administration, counsel, education and marketing. While not all guidelines mandate providers promote vaccination to clients, we did not locate any that expressly prohibit vaccine promotion or permit any counsel that may be seen as "anti-vaccination". However, one could reasonably expect the ambiguity in many policy statements makes them liable to being interpreted and acted upon in different ways. For instance, the College of Naturopaths of Ontario [42] requires providers to "encourage the patient to be an active participant in his/her own health care, which allows the patient to make fully informed decisions," but does not specify which types of information can be presented to patients.

With the exception of the Ontario Association of Naturopathic Doctors [44], the Manitoba Homeopathic Association [46] and the BC Association of Homeopaths [47], none of the other 45 reviewed webpages made any claims themselves or referred to other sources that emphasize the alleged harmfulness or ineffectiveness of vaccines. This is a striking departure from the peer-reviewed literature, where claims of this sort were commonplace. We cannot offer a definitive explanation for this discrepancy without further information, but we speculate this could be a result of geographic differences in professional norms between Canadian organizations and the academic literature, which originates mostly from the United States. Alternatively, displaying mostly pro-vaccination attitudes on public webpages could be part of a strategy to "save face" among the general public and key stakeholders while practitioners debate vaccine safety and effectiveness within the confines of paywalled (restricted access via paid subscription) academic journals. Examples abound in Canadian popular media of CAM providers being criticized [62, 63] and promptly subject to disciplinary action [64, 65] after publicly opposing vaccination. Vaccine hesitancy has also been shown to be heavily stigmatized by the general public [66, 67].

It is important that we clarify some of the limits to the conclusions that may be drawn from the current study. First, we wish to reiterate that any points made with regard to vaccine attitudes in the CAM literature and organizational webpages are not intended to be generalizable to chiropractors, naturopaths and homeopaths by and large. Although this review qualitatively describes professional discourses that *may potentially* influence practitioners' vaccine-related knowledge, attitudes and practices, it is outside the scope of this study to determine precisely

how and to what extent these socio-professional influences are actually reflected among practitioners "in the field". Previous research demonstrating a significant presence of providers both supportive and critical of vaccination within CAM professions would suggest that any of the vaccine-related views and standards of practice detailed herein likely elicit only partial support or compliance [68, 69]. It is also extraneous to our study objectives to evaluate the veracity of the claims made within the reviewed works; here, our goal is descriptive. A systematic evaluation of the type and quality of evidence CAM providers use to support their vaccine-related arguments is a compelling direction for future inquiry. Similarly, an investigation into how the arguments and prescriptions of chiropractic, naturopathic and homeopathic authors and organizations compare to their counterparts in evidence-based medicine would offer important contextual knowledge, but ultimately was not part of this work.

Some limitations inherent to our chosen methods also warrant acknowledgement. Braun and Clarke [32] note that thematic analysis involves searching *across* a dataset to identify repeat patterns of meaning, and as such, it is less suited to uncovering continuities and contradictions *within* any one data item compared to narrative or biographical approaches. Though we could glean from coding that certain sub-themes clustered around particular sources, it would require a separate form of analysis to uncover how these themes relate to one another and together form a coherent position on vaccination for any one source. Moreover, in contrast to more theoretically- or epistemologically-bounded methods, like discourse analysis, simple thematic analysis does not permit us to explore how language is used and to what effect [32]. Insofar as vaccine-related views are communicated through means beyond the literal interpretation of texts, we are unable to evaluate this using our current methods. However, since our aim was to describe and not interpret CAM writings, we feel our analytic approach that attended only to manifest themes was the most appropriate. Finally, our choice to provide a detailed account of one particular aspect of the dataset–focusing specifically on claims made in explicit reference to vaccination–instead of offering a rich description of the entire dataset means that some analytic breadth was sacrificed for the sake of depth. A variety of comments were made regarding government, public health institutions, health practitioners and biomedical onto-epistemology that on the surface did not seem to pertain directly to vaccination, and as a result fall outside the scope of our analysis. It has been shown, however, that vaccine hesitancy often reflects deeper concerns about medicine, the state and the body and a growing distrust of health professionals, the pharmaceutical industry and government [70, 71]. It remains unclear if and how these views are relevant to vaccination attitudes expressed by the authors surveyed in the current study.

We also wish to direct readers' attention to articles in S3 Table for which we note are "undetermined" with regards to whether they are sourced from a reputable, peer-reviewed journal. At the direction of a library technician, journals' peer-reviewed status were verified by their according classification in Ulrich's Periodicals Directory and EBSCO. All included journals were listed as peer-reviewed in both databases with two exceptions: Homoeopathic Links and Homoeopathic Heritage. Homoeopathic Links was listed as peer-reviewed/refereed by EBSCO but not Ulrich's. On the journal's webpage (https://www.thieme.in/homoeopathic-links), the author instructions note the journal uses single-blind peer review, but offers no further details of the review process. Homoeopathic Heritage, inversely, is listed as peer-reviewed/refereed in Ulrich's but not EBSCO. The journal's webpage (https://www.bjainbooks.com/inr/journal.html/) provides a definition of peer review but does not explicitly state that it uses peer review in the publication process. We opted to include articles from these journals despite the ambiguity in their peer-reviewed status given that in doing so we may omit papers that met our inclusion criteria. However, we submit that any claims made in articles originating from these sources should be considered in light of this ambiguity.

Limitations notwithstanding, the current study marks one of the first attempts at characterizing the vaccine-related positions of key institutions that directly influence CAM practice, including the research community, professional organizations and regulatory bodies. As these organizations further evolve in Canada and elsewhere as part of a broader "professionalization" initiative in CAM [72], greater attention will need to be directed at their role in shaping CAM providers' beliefs and practices that both support and undermine public health objectives. Our hope is this and future studies on vaccine hesitancy in CAM facilitate constructive dialogue between researchers, public health experts, policymakers, mainstream healthcare providers and CAM providers to more closely align our objectives and better coordinate vaccine promotion efforts.

## Supporting information

**S1 Table. Database search strategies and results for scoping review of peer-reviewed literature.**
(DOCX)

**S2 Table. Characteristics of sources retrieved from grey literature scan of professional organization webpages.**
(DOCX)

**S3 Table. Characteristics of articles retrieved from scoping review of peer-reviewed literature.**
(DOCX)

**S1 Checklist.**
(PDF)

**S1 Fig.**
(TIF)

## Acknowledgments

We would like to thank Jackie Stapleton BSc, MLS for aiding the development of the search strategy for this review.

## Author Contributions

**Conceptualization:** Eve Dubé, Janice E. Graham, Noni E. MacDonald, Julie A. Bettinger, Devon Greyson, Shannon MacDonald, S. Michelle Driedger, Samantha B. Meyer.

**Data curation:** Eric Filice, Eve Dubé.

**Formal analysis:** Eric Filice.

**Funding acquisition:** Eve Dubé, Janice E. Graham, Noni E. MacDonald, Julie A. Bettinger, Devon Greyson, Shannon MacDonald, S. Michelle Driedger, Samantha B. Meyer.

**Investigation:** Eric Filice.

**Methodology:** Eric Filice, Eve Dubé, Janice E. Graham, Noni E. MacDonald, Julie A. Bettinger, Devon Greyson, Shannon MacDonald, S. Michelle Driedger, Samantha B. Meyer.

**Supervision:** Eve Dubé, Samantha B. Meyer.

**Writing – original draft:** Eric Filice.

**Writing – review & editing:** Eric Filice, Eve Dubé, Janice E. Graham, Noni E. MacDonald, Julie A. Bettinger, Devon Greyson, Shannon MacDonald, S. Michelle Driedger, Greg Kawchuk, Samantha B. Meyer.

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
