## [Decision Letter · Decision Letter 0]

28 May 2020

PONE-D-20-05690

Vaccination discourses among chiropractors, naturopaths and homeopaths: a qualitative content analysis of academic literature and Canadian organizational webpages

PLOS ONE

Dear Dr. Meyer,

Thank you for submitting your manuscript to PLOS ONE. After careful consideration, we feel that it has merit but does not fully meet PLOS ONE’s publication criteria as it currently stands. Therefore, we invite you to submit a revised version of the manuscript that addresses the points raised during the review process.

We look forward to receiving your revised manuscript.

Kind regards,

M Barton Laws

Academic Editor

PLOS ONE

Additional Editor Comments:

Reviewer 1 has made some comments which are debatable. You do not exclusively cite opinions unfavorable to vaccination, and you do separate your discussion of academic literature from other sources. Nevertheless you should consider why the reviewer responded in this way and make the rationale for what you have done more clear. The criticism of one journal that you cite in the discussion is legitimate but you should more fully describe the source, as would be helpful to the reader in many other instances. I believe there is substantial controversy about the standards of some CAM journals and their standards of evidence, which you might want to explore more deeply. While I understand that your purpose is to present what you find out in the world, and not criticism, I don't think you need to be entirely agnostic about the quality of evidence in the academic, or in some cases questionably academic literature that you review. You state that a full inquiry into the quality of this evidence is a fit subject for future inquiry but you could better emphasize the importance of the question. Also, please pay attention to Reviewer 2's comments. What appears on the web sites of the associations is not necessarily informative about how CAM practitioners actually advise clients. What you find in the published literature suggests to me that there is more skepticism among practitioners than the official guidance would indicate. Reviewer 1 also refers to controversy among CAM advocates and practitioners, which you might want to describe more fully.

I agree that the term "allopathic" medicine is questionable. This term arose historically to describe a school of medicine in opposition to homeopathy. However, that is no longer definitional of what is taught in medical school and ascribes too much status to homeopathy as an equally plausible alternative. It would be better to say science based medicine, or perhaps standard medical practice. Finally, I agree with both reviewers that you should try to shorten this considerably.

Journal Requirements:

Reviewers' comments:

Reviewer's Responses to Questions

**Comments to the Author**

1. Is the manuscript technically sound, and do the data support the conclusions?

Reviewer #1: No

Reviewer #2: Yes

2. Has the statistical analysis been performed appropriately and rigorously? 

Reviewer #1: N/A

Reviewer #2: Yes

3. Have the authors made all data underlying the findings in their manuscript fully available?

Reviewer #1: Yes

Reviewer #2: Yes

4. Is the manuscript presented in an intelligible fashion and written in standard English?

Reviewer #1: Yes

Reviewer #2: Yes

5. Review Comments to the Author

Reviewer #1: There is too much heterogeneity among sources to grant joint analysis: papers in peer-reviewed, PubMed and JCR indexed journals are analysed together with blogs, opinions, and even comments by non-practitioners!

It's not clear why several papers were included for analysis, nor why you "randomly selected 30 from 65".

There is bias in the presentation of the results: only opinions against vaccines were described, and you fully omitted discussions within communities of practitioners. In addition to "n" you should provide the reference for the citations, so readers can know how many of the included papers are actually cited.

The attempt to disqualify journal Homeopathy (peer-reviewed, PubMed and JCR indexed) using a blog (!) is uncalled for and makes explicit the bias that impregnates the entire Results and large part of the Discussion section/

Reviewer #2: Interesting and well written paper. I have two concerns. The paper is rather long, and it would be much better if the authors try to make it shorter. The second one has already been mentioned in the manuscript. “It is also extraneous to our study objectives to evaluate the veracity of the claims made within the reviewed works; here, our goal is descriptive. A systematic evaluation of the type and quality of evidence CAM providers use to inform their vaccine-related arguments is a compelling direction for future inquiry.”. We all know that despite the fact that CAM physicians do not have any guidelines against vaccines their attitude and practice is different. One of the results in our recent study (Vaccine confidence among parents: Large scale study in eighteen European countries) published in the vaccine journal is the following: Parents who consult with paediatricians or nurses had lower hesitancy scores than parents who consult either with GPs (15.4 vs. 23.1, p < 0.05) or with homeopathists (15.4 vs. 51.3, p < 0.05). So, we need to be very careful not to give the wrong message to the reader.

6. PLOS authors have the option to publish the peer review history of their article (what does this mean?). If published, this will include your full peer review and any attached files.

Reviewer #1: No

Reviewer #2: Yes: ADAMOS HADJIPANAYIS

---

## [Author Response · Author response to Decision Letter 0]

10 Jun 2020

We have uploaded a table in our resubmission that provides detailed responses to each concern raised by the editor and reviewers. Thank you.

---

## [Editor Report · Decision Letter 1]

14 Jul 2020

Vaccination discourses among chiropractors, naturopaths and homeopaths: a qualitative content analysis of academic literature and Canadian organizational webpages

PONE-D-20-05690R1

Dear Dr. Meyer,

We’re pleased to inform you that your manuscript has been judged scientifically suitable for publication and will be formally accepted for publication once it meets all outstanding technical requirements.

Kind regards,

M Barton Laws

Academic Editor

PLOS ONE

Journal Requirements:

In the Methods, or in Figure 1, please include the reason(s) why 19 full-text articles were excluded during the screening process after de-duplication and before analysis.

Additional Editor Comments (optional):

I believe you have satisfactorily addressed the reviewers' concerns and my own.
---

## [Editor Report · Acceptance letter]

24 Jul 2020

PONE-D-20-05690R1 

Vaccination discourses among chiropractors, naturopaths and homeopaths: a qualitative content analysis of academic literature and Canadian organizational webpages 

Dear Dr. Meyer:

I'm pleased to inform you that your manuscript has been deemed suitable for publication in PLOS ONE. Congratulations! Your manuscript is now with our production department. 

Kind regards, 

on behalf of

Dr. M Barton Laws 

Academic Editor

PLOS ONE